# Towards the use of conservative thermodynamic variables in data assimilation: a case study using ground-based microwave radiometer measurements

Pascal Marquet[1], Pauline Martinet[1], Jean-François Mahfouf[1], Alina Lavinia Barbu[1], and Benjamin Ménétrier[2]

[1]CNRM, Université de Toulouse, Météo-France, CNRS, Toulouse, France
[2]INP, IRIT, Université de Toulouse, Toulouse, France

**Correspondence:** Pascal Marquet (pascal.marquet@meteo.fr, pascalmarquet@yahoo.com) Pauline Martinet (pauline.martinet@meteo.fr)

**Abstract.** This study aims at introducing two conservative thermodynamic variables (moist-air entropy potential temperature and total water content) into a one-dimensional variational data assimilation system (1D-Var) to demonstrate the benefit for future operational assimilation schemes. This system is assessed using microwave brightness temperatures from a ground-based radiometer installed during the field campaign SOFOG3D dedicated to fog forecast improvement.

5    An underlying objective is to ease the specification of background error covariance matrices that are highly dependent on weather conditions when using classical variables, making difficult the optimal retrievals of cloud and thermodynamic properties during fog conditions. Background error covariance matrices for these new conservative variables have thus been computed by an ensemble approach based on the French convective scale model AROME, for both all-weather and fog conditions. A first result shows that the use of these matrices for the new variables reduces some dependencies to the meteorological conditions (diurnal cycle, presence or not of clouds) compared to usual variables (temperature, specific humidity).

Then, two 1D-Var experiments (classical vs. conservative variables) are evaluated over a full diurnal cycle characterized by a stratus-evolving radiative fog situation, using hourly brightness temperatures.

Results show, as expected, that analysed brightness temperatures by the 1D-Var are much closer to the observed ones than background values for both variable choices. This is especially the case for channels sensitive to water vapour and liquid water.

15   On the other hand, analysis increments in model space (water vapour, liquid water) show significant differences between the two sets of variables.

## 1   Introduction

Numerical Weather Prediction (NWP) models at convective scale need accurate initial conditions for skillful forecasts of high impact meteorological events taking place at small-scale such as convective storms, wind gusts or fog. Observing systems

20   sampling atmospheric phenomena at small-scale and high temporal frequency are thus necessary for that purpose (Gustafsson et al., 2018). Ground-based remote-sensing instruments (e.g. rain and cloud radars, radiometers, wind profilers) meet such

requirements and provide information on wind, temperature, and atmospheric water (vapour and hydrometeors). Moreover, data assimilation systems are evolving towards ensemble approaches where hydrometeors can be initialized together with usual control variables. This is the case for the Météo-France NWP limited area model AROME (Seity et al., 2011; Brousseau et al., 2016) where, on top of wind $(U, V)$, temperature $(T)$ and specific humidity $q_v$, several hydrometeor mass contents can be initialized (cloud liquid water $q_l$, cloud ice water $q_i$, rain $q_r$, snow $q_s$ and graupel $q_g$) (Destouches et al., 2021). However, these variables are not conserved during adiabatic and reversible vertical motion.

The accuracy of the analysed state in variational schemes highly depends on the specification of the so-called background error covariance matrix. Background error variances and cross-correlations between variables are known to be dependent on weather conditions (Montmerle and Berre, 2010; Michel et al., 2011). This is particularly the case during fog conditions with much shorter vertical correlation length-scales at the lowest levels and large positive cross-correlations between temperature and specific humidity (Ménétrier and Montmerle, 2011). In this context, Martinet et al. (2020) have demonstrated that humidity retrievals could be significantly degraded if sub-optimal background error covariances are used during the minimization. New ensemble approaches allow a better approximation of background error covariance matrices but rely on the capability of the ensemble data assimilation to correctly represent model errors, which might not always be the case during fog conditions. This is why it would be of interest to examine, in a data assimilation context, the use of variables that are more suitable when water phase changes take place.

It is well-known that most data assimilation systems were based on the assumptions of homogeneity and isotropy of background error correlations. To release these hypotheses, Desroziers and Lafore (1993) and Desroziers (1997) implemented a coordinate change inspired by the semi-geostrophic theory to test flow-dependent analyses with case studies from the Front-87 field campaign (Clough and Testud, 1988), where the local horizontal coordinates were transformed into the semi-geostrophic space during the assimilation process. Another kind of flow-dependent analyses were made by Cullen (2003) and Wlasak et al. (2006) who proposed a low-order Potential Vorticity (PV) inversion scheme to define a new set of control variables. Similarly, analyses on potential temperature $\theta$ were made by Shapiro and Hastings (1973) and Benjamin et al. (1991), and more recently by Benjamin et al. (2004) with moist virtual $\theta_v$ and moist equivalent $\theta_e$ potential temperatures.

The aim of the paper is to test a one-dimensional data assimilation method that would be less sensitive to the average vertical gradients of the $(T, q_v, q_l, q_i)$ variables. To this end, two conservative variables will be proposed, generalizing previous uses of $\theta$ (as a proxy for the entropy of dry air) to moist-air variables suitable for data assimilation. The new conservative variables are the total water content $q_t = q_v + q_l + q_i$ and the moist-air entropy potential temperature $\theta_s$ defined in Marquet (2011), which generalize the two well-known conservative variables $(q_t, \theta_l)$ of Betts (1973).

The focus of the study will be on a fog situation from the SOFOG3D field campaign using a one-dimensional variational (1D-Var) system for the assimilation of observed microwave brightness temperatures sensitive to $T$, $q_v$ and $q_l$ from a ground-based radiometer. Short-range forecasts from the convective scale model AROME (Seity et al., 2011) will be used as background profiles, the fast radiative transfer model RTTOV-gb (De Angelis et al., 2016; Cimini et al., 2019) will allow to accurately simulate the brightness temperatures, and suitable background error covariance matrices will be derived from an ensemble technique.

Section 2 presents the methodology (conservative variables, 1D-Var, change of variables). Section 3 describes the experimental setting, the meteorological context, the observations and the different components of the 1D-Var system. The results are commented in Section 4. Finally, conclusions and perspectives are given in Section 5.

## 2 The methods

This section presents the methodology chosen for this study. The definition of the moist-air entropy potential temperature $\theta_s$ is introduced, as well as the formalism of the 1D-Var assimilation system, before describing the "conservative variable" conversion operator.

### 2.1 The moist-air entropy potential temperature

The motivation for using the absolute moist-air entropy in atmospheric science was first described by Richardson (1919, 1922), and then fully formalized by Hauf and Höller (1987). The method is to take into account the absolute value for dry air and water vapour and to define a moist-air entropy potential temperature variable called $\theta_s$.

However, the version of $\theta_s$ published in Hauf and Höller (1987) was not really synonymous with the moist-air entropy. This problem has been solved with the version of Marquet (2011) by imposing the same link with the specific entropy of moist air ($s$) as in the dry-air formalism of Bauer (1908), leading to:

$$s = c_{pd} \ln\left(\frac{\theta_s}{T_0}\right) + s_{d0}(T_0, p_0), \tag{1}$$

where $c_{pd} \approx 1004.7\,\mathrm{J\,K^{-1}\,kg^{-1}}$ is the dry-air specific heat at constant pressure, $T_0 = 273.15\,\mathrm{K}$ is a standard temperature and $s_{d0}(T_0, p_0) \approx 6775\,\mathrm{J\,K^{-1}\,kg^{-1}}$ is the reference dry-air entropy at $T_0$ and at the standard pressure $p_0 = 1000\,\mathrm{hPa}$. Because $c_{pd}$, $T_0$, and $s_{d0}(T_0, p_0)$ are constant terms, $\theta_s$ defined by (1) is synonymous with, and has the same physical properties as, the moist-air entropy $s$.

The conservative aspects of this potential temperature $\theta_s$ and its meteorological properties (in e.g. fronts, convection, cyclones) have been studied in Marquet (2011), Blot (2013) and Marquet and Geleyn (2015). The links with the definition of the Brunt-Väisälä frequency and the potential vorticity are described in Marquet and Geleyn (2013) and Marquet (2014), while the significance of the absolute entropy to describe the thermodynamics of cyclones is shown in Marquet (2017) and Marquet and Dauhut (2018).

Only the first order approximation of $\theta_s$, noted $(\theta_s)_1$ in Marquet (2011), will be considered in the following, that writes:

$$\theta_s \approx (\theta_s)_1 = \theta \exp\left(-\frac{L_{vap}\,q_l + L_{sub}\,q_i}{c_{pd}\,T}\right) \exp(\Lambda_r\,q_t), \tag{2}$$

where $\theta = T\,(p/p_0)^\kappa$ is the dry-air potential temperature, $p$ the pressure, $\kappa \approx 0.2857$, $L_{vap}(T)$ and $L_{sub}(T)$ the latent heat of vaporization and sublimation, respectively. The explanation for $\Lambda_r$ follows later in the section.

The first term $\theta$ on the right-hand side of (2) leads to a first conservation law (invariance) during adiabatic compression and expansion, with joint and opposite variations of $T$ and $p$ keeping $\theta$ constant. Here lies the motivation for using $\theta$ to describe dry-air convective processes, and also in data assimilation systems by Shapiro and Hastings (1973) and Benjamin et al. (1991).

The first exponential on the right-hand side of (2) explains a second form of conservation law. Indeed, this exponential is constant for reversible and adiabatic phase changes, for which $d(c_{pd}\,T) \approx d(L_{vap}\,q_l + L_{sub}\,q_i)$ due to the approximate conser-

vation of the moist-static energy $c_{pd}\,T - L_{vap}\,q_l - L_{sub}\,q_i$, and with therefore joint variations of the numerator and denominator and a constant fraction into the first exponential. It should be mentioned that the product of $\theta$ by this first exponential forms the Betts (1973) conservative variable $\theta_l$, which is presently used together with $q_t$ to describe the moist-air turbulence in GCM and NWP models.

While the variable $\theta_l$ was established with the assumption of a constant total water content $q_t$ in Betts (1973), the second

exponential in (2) sheds new light on a third and new conservation law, where the entropy of moist air can remain constant despite changes in the total water $q_t$. This occurs in regions where water vapour turbulence transport takes place, or via the evaporation process over oceans, or at the edges of clouds via entrainment and detrainment processes.

We consider here "open-system" thermodynamic processes, for which the second exponential takes into account the impact on moist-air entropy when the changes in specific content of water vapour are balanced, numerically, by opposite changes of

dry air, namely with $dq_d = -\,dq_t \neq 0$. In this case, as stated in Marquet (2011), the changes in moist-air entropy depend on reference values (with subscript "r") according to $d[\,q_d\,(s_d)_r + q_t\,(s_v)_r\,]$, and thus with $(s_d)_r$ and $(s_v)_r$ being constant and with the relation $q_d = 1 - q_t$, leading to $[\,(s_v)_r - (s_d)_r\,]\,dq_t$.

This explains the new term $\Lambda_r = [\,(s_v)_r - (s_d)_r\,]/c_{pd} \approx 5.869 \pm 0.003$, which depends on the absolute reference entropies for water vapour $(s_v)_r \approx 12671\ \mathrm{J\,K^{-1}\,kg^{-1}}$ and dry air $(s_d)_r \approx 6777\ \mathrm{J\,K^{-1}\,kg^{-1}}$. This also explains that these "open-system"

thermodynamic effects can be taken into account to highlight regimes where the specific moist-air entropy $(s)$, $\theta_s$ and $(\theta_s)_1$ can be constant despite changes in $q_t$, which may decrease or increase on the vertical (see Marquet, 2011, for such examples).

Although it should be possible to use $(\theta_s)_1$ as a control variable for assimilation, it appeared desirable to define an additional approximation of this variable for a more "regular" and more "linear" formulation, insofar as tangent-linear and adjoint versions are needed for the 1D-Var system. Considering the approximation $\exp(x) \approx 1+x$ for the two exponentials in (2), neglecting the

second order terms in $x^2$, also neglecting the variations of $L_v(T)$ with temperature and assuming a no-ice hypothesis ($q_i = 0$), the new variable writes:

$$(\theta_s)_a = \theta\left[1 + \Lambda_r\,q_t - \frac{L_{vap}(T_0)\,q_l}{c_{pd}\,T}\right], \tag{3}$$

$$(\theta_s)_a = \frac{1}{c_{pd}}\left(\frac{p_0}{p}\right)^{\kappa}\left[\,c_{pd}\,(1 + \Lambda_r\,q_t)\,T - L_{vap}(T_0)\,q_l\,\right], \tag{4}$$

where $L_{vap}(T_0) \approx 2501\ \mathrm{kJ\,kg^{-1}}$. This formulation corresponds to $S_m/c_{pd}$, where $S_m$ is the Moist Static Energy defined in

Marquet (2011, Eq. 73) and used in the ECMWF[1] NWP global model by Marquet and Bechtold (2020).

---

[1] European Centre for Medium range Weather Forecasts

The new potential temperature $(\theta_s)_a$ remains close to $(\theta_s)_1$ (not shown) and keeps almost the same three conservative properties described for $(\theta_s)_1$. This new conservative variable $(\theta_s)_a$ will be used along with the total water content $q_t = q_v + q_l$ in the data assimilation experimental context described in the following sections.

## 2.2 The 1D-Var formalism

The general framework describing the retrieval of atmospheric profiles from remote-sensing instruments by statistical methods can be found in Rodgers (1976). In the following we present the main equations of the one-dimensional variational formalism. Additional details are given in Thépaut and Moll (1990) who developed the first 1D-Var inversion applied to satellite radiances using the adjoint technique.

The 1D-Var data assimilation system searches for an optimal state (the analysis) as an approximate solution of the problem

minimizing a cost function $\mathcal{J}$ defined by:

$$
\begin{aligned}
\mathcal{J}(x) = & \frac{1}{2} \left( x - x_b \right)^T \mathbf{B_x}^{-1} \left( x - x_b \right) \\
& + \frac{1}{2} \left[ y - \mathcal{H}(x) \right]^T \mathbf{R}^{-1} \left[ y - \mathcal{H}(x) \right].
\end{aligned}
\tag{5}
$$

The symbol $^T$ represents the transpose of a matrix.

The first (background) term measures the distance in model space between a control vector $x$ (in our study, $T$, $q_v$ and $q_l$

profiles) and a background vector $x_b$, weighted by the inverse of the background error covariance matrix ($\mathbf{B_x}$) associated with the vector $x$. The second (observation) term measures the distance in the observation space between the value simulated from the model variables $\mathcal{H}(x)$ (in our study, the radiative transfer model RTTOV-gb) and the observation vector $y$ (in our study, a set of microwave brightness temperatures from a ground-based radiometer), weighted by the inverse of the observation error covariance matrix ($\mathbf{R}$). The solution is searched iteratively by performing several evaluations of $\mathcal{J}$ and its gradient:

$$
\nabla_x \mathcal{J}(x) = \mathbf{B_x}^{-1} \left( x - x_b \right) - \mathbf{H}^T \mathbf{R}^{-1} \left[ y - \mathcal{H}(x) \right],
\tag{6}
$$

where $\mathbf{H}$ is the Jacobian matrix of the observation operator representing the sensitivity of the observation operator to changes in the control vector $x$ ($\mathbf{H}^T$ is also called the adjoint of the observation operator).

## 2.3 The conversion operator

The 1D-Var assimilation defined previously with the variables $x = (T, q_v, q_l)$ can be modified to use the conservative variables

$z = ((\theta_s)_a, q_t)$. A conversion operator that projects the state vector from one space to the other can be written as $x = \mathcal{L}(z)$. In the presence of liquid water $q_l$, an adjustment to saturation is made to separate its contribution to the total water content $q_t$ from the water vapour content $q_v$. This is equivalent to distinguishing the "unsaturated" case from the "saturated" one. Therefore, starting from initial conditions $(T_I, q_I) = (T, q_v)$ and using the conservation of $(\theta_s)_a$ given by Eq. (4), we look for the variable

$T^*$ such that:

$$T^* + \alpha\, q_{sat}(T^*) = T_I + \alpha\, q_I, \tag{7}$$

$$\text{where } \alpha = \frac{L_{vap}(T_0)}{c_{pd}\,(1 + \Lambda_r\, q_t)} \tag{8}$$

and $q_{sat}(T^*)$ is the specific humidity at saturation.

For the unsaturated case ($q_v < q_{sat}(T^*)$), we obtain the variables $(T, q_v, q_l)$ directly from Eq. (4):

$$q_l = 0\,, \;\; q_v = q_t \;\; \text{and} \;\; T = (\theta_s)_a \left(\frac{p}{p_0}\right)^\kappa \frac{1}{1 + \Lambda_r\, q_t}. \tag{9}$$

For the saturated case ($q_v \geq q_{sat}(T^*)$), we write:

$$q_l = q_t - q_{sat}(T^*) \;\; \text{and} \;\; q_v = q_{sat}(T^*)\,. \tag{10}$$

In this situation, it is necessary to calculate implicitly the temperature $T^*$, given by Equation (7). We compute numerically an approximation of $T^*$ by an iterative Newton's algorithm.

Taking into account this change of variables, the cost-function can be written as:

$$\mathcal{J}(z) = \frac{1}{2}\,(z - z_b)^T\,\mathbf{B_z}^{-1}\,(z - z_b)$$
$$+ \frac{1}{2}\,[\,y - \mathcal{H}(\mathcal{L}(z))\,]^T\,\mathbf{R}^{-1}\,[\,y - \mathcal{H}(\mathcal{L}(z))\,]\,. \tag{11}$$

Then, its gradient given by Eq. (6) becomes:

$$\nabla_z \mathcal{J}(z) = \mathbf{B_z}^{-1}(z - z_b) - \mathbf{L}^T \mathbf{H}^T \mathbf{R}^{-1}[y - \mathcal{H}(\mathcal{L}(z))], \tag{12}$$

where $\mathbf{L}^T$ is the adjoint of the conversion operator $\mathcal{L}$.

The second term on the right hand side of Eq. (11) and (12) indicates that the conversion operator $\mathcal{L}$ is needed to compute the brightness temperatures from the observation operator $\mathcal{H}$. Indeed RTTOV-gb requires profiles of temperature, specific humidity and liquid water content as input quantities. This space change is required at each step of the minimisation process. For the computation of the gradient of the cost-function $\nabla_z \mathcal{J}$, the linearized version (adjoint) of $\mathcal{L}$ is also necessary. In practice, the operator $\mathbf{L}^T$ provides the gradient of the brightness temperatures with respect to the conservative variables, knowing the
gradient with respect to the classical variables.

## 3  The experimental setup

The numerical experiments to be presented afterwards will use measurements made during the SOFOG3D field experiment[2] (SOuth west FOGs 3D experiment for processes study) that took place from 1 October 2019 to 31 March 2020 in southwestern France to advance understanding of small-scale processes and surface heterogeneities leading to fog formation and dissipation.

---

[2]https://www.umr-cnrm.fr/spip.php?article1086

Many instruments were located at the Saint-Symphorien super-site (Les Landes region), such as a HATPRO (Humidity and Temperature PROfiler, Rose et al., 2005), a 95 GHz BASTA Doppler cloud radar (Delanoë et al., 2016), a Doppler lidar, an aerosol lidar, a surface weather station and a radiosonde station. One objective of this campaign was to test the contribution of the assimilation of such instrumentation on the forecast of fog events by NWP models.

### 3.1 The 9 February 2020 situation

This section presents the experimental context of 9 February 2020 at the Saint-Symphorien site characterized by (i) a radiative fog event observed in the morning and (ii) the development of low-level clouds in the afternoon and evening.

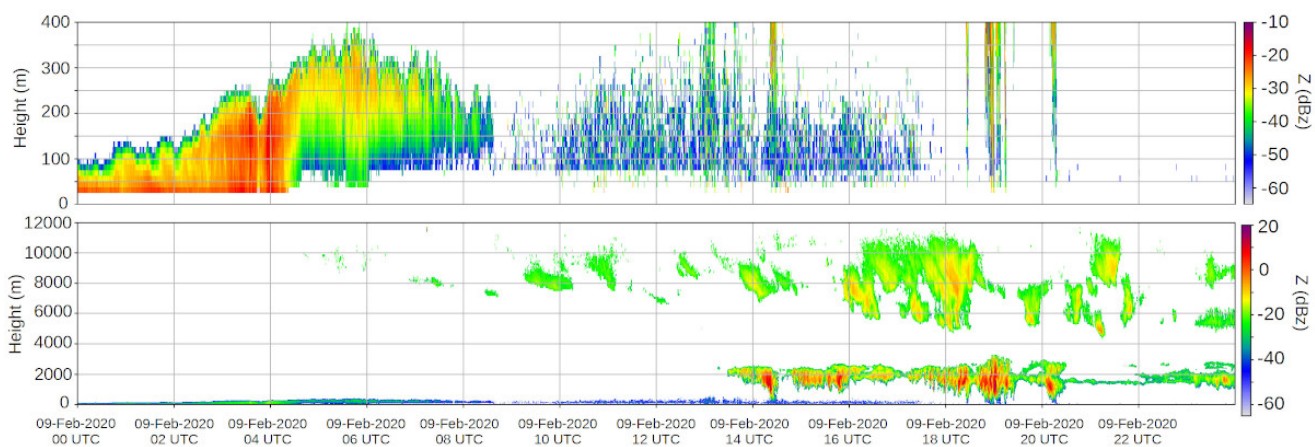

**Figure 1.** Reflectivity profiles at 95 GHz (dBZ) measured by the BASTA cloud radar in the first 500 m (top) and up to 12000 m altitude (bottom), with UTC hours in abscissas, for the day of 9 February 2020 at Saint-Symphorien (Les Landes region). From http://basta.projet. latmos.ipsl.fr/?bi=bif

Figure 1 shows a time series of cloud radar reflectivity profiles (W-band at 95 GHz) measured by the BASTA instrument (Delanoë et al., 2016) in the lowest hundred meters (top panel) between 9 February 2020 at 00 UTC and 10 February 2020 at 00 UTC. The instrument reveals a thickening of the fog between midnight and 04 UTC. The fog layer thickness is located

between 90 m and 250 m. After 04 UTC, the fog layer near the ground rises, lifting in a "stratus" type cloud (between 100 and 300 m). After 08 UTC, the stratus cloud dissipates. In the bottom panel BASTA observations up to 12000 m ($\approx$ 200 hPa) indicate low-level clouds after 14 UTC, generally between 1000 m ($\approx$ 900 hPa) and 2000 m ($\approx$ 780 hPa), with a fairly good agreement with AROME short-range (1 h) forecasts (see Fig. 2 (f)). Optically thin (reflectivity below 0 dBZ) high altitude ice clouds are also captured by the radar.

Figure 2 depicts the diurnal cycle evolution in terms of vertical profiles of: (a) absolute temperature $T$, (b) dry-air potential temperature $\theta$, (c) water vapour specific content $q_v$, (d) entropy potential temperature $(\theta_s)_a$, (e) cloud liquid water specific content $q_l$ and (f) relative humidity ($RH$), from 1 h AROME forecasts (background) of 9 February 2020 at Saint-Symphorien.

**Table 1.** Channel numbers, band frequencies (GHz) and observation uncertainties (K) prescribed in the observation-error-covariance matrix (from Martinet et al., 2020)

| Channel numbers | 1 | 2 | X | 4 | 5 | 6 | 7 |
|---|---|---|---|---|---|---|---|
| K-band Frequencies (GHz) | 22.24 | 23.04 | X | 25.44 | 26.24 | 27.84 | 31.4 |
| K-band $\sigma_o(K)$ | 1.34 | 1.71 | X | 1.08 | 1.25 | 1.17 | 1.19 |
| Channel numbers | 8 | 9 | 10 | 11 | 12 | 13 | 14 |
| V-band Frequencies (GHz) | 51.26 | 52.28 | 53.86 | 54.94 | 56.66 | 57.3 | 58 |
| V-band $\sigma_o(K)$ | 3.21 | 3.29 | 1.30 | 0.37 | 0.42 | 0.42 | 0.36 |

At this stage, it is important to indicate that the AROME model has a 90-level vertical discretisation from the surface up to 10 hPa, with a high resolution in the Planetary Boundary Layer (PBL) since 20 levels are below 2 km.

Figures 2 (e) and (f), for $q_l$ and RH, show two main saturated layers: a fog layer close to the surface between 00 and 09 UTC with the presence of a thin liquid cloud layer aloft at $850$ hPa at 00 UTC, and the presence of a stratocumulus cloud between 14 UTC and midnight at $850$ hPa. During the night, near surface layers cool down, with a thermal inversion that sets at around 01 UTC and persists until 07 UTC. After the transition period between 06 UTC and 09 UTC, when the dissipation of the fog and stratus takes place, the air warms up and the PBL develops vertically (see the black curves plotted where vertical gradients of $\theta$ in (b) are large). Towards the end of the day, the thickness of the PBL remains important until 24 UTC, probably due to the presence of clouds between $800$ and $750$ hPa that reduces the radiative cooling (see Figs. 2 (c) and (f) for $q_v$ and RH).

Figure 2(d) reveals weaker vertical gradients for the $(\theta_s)_a$ profiles, notably with contour lines often vertical and less numerous than those of the $T$, $\theta$ and $q_v$ profiles in (a), (b) and (c), as also shown by more extensive and more numerous vertical arrows in (d) than in (b). Here we see the impact of the coefficient $\Lambda_r \approx 5.869$ in Eqs. (3)-(4), which allows the vertical gradients of $\theta$ in (b) and $q_v$ in (c) to often compensate each other in the formula for $(\theta_s)_a$. This is especially true between $980$ hPa and $750$ hPa in the morning between 04 and 10 UTC, and also within the dry and moist boundary layers during the day.

Note that the dissipation of the fog is associated with a homogenization of $(\theta_s)_a$ in (d) from 04 to 05 UTC in the whole layer above, in the same way as the transition from strato-cumulus toward cumulus was associated with a cancellation of the vertical gradient of $(\theta_s)_1$ in the Fig. 6 of Marquet and Geleyn (2015). This phenomenon cannot be easily deduced from the separate analysis of the gradients of $\theta$ and $q_v$ in (b) and (c). Therefore, three air mass changes can be clearly distinguished during the day. The vertical gradients of $(\theta_s)_a$ are stronger during cloudy situations, first (i) at night and early morning before 04 UTC and just above the fog, then (ii) at the end of the day above the top-cloud level at $800$ hPa; with (iii) turbulence-related phenomena in between that mix the air mass and $(\theta_s)_a$, up to the cloudy layer tops that evolve between $950$ and $800$ hPa from 13 UTC to 17 UTC.

The observations to be assimilated are presented in the following. The HATPRO MicroWave Radiometer (MWR) measures brightness temperatures ($TB$) at 14 frequencies (Rose et al., 2005) between 22.24 and 58 GHz: 7 are located in the water vapour absorption K-band and 7 are located in the oxygen absorption V-band (see the Table 1). For our study, the third channel

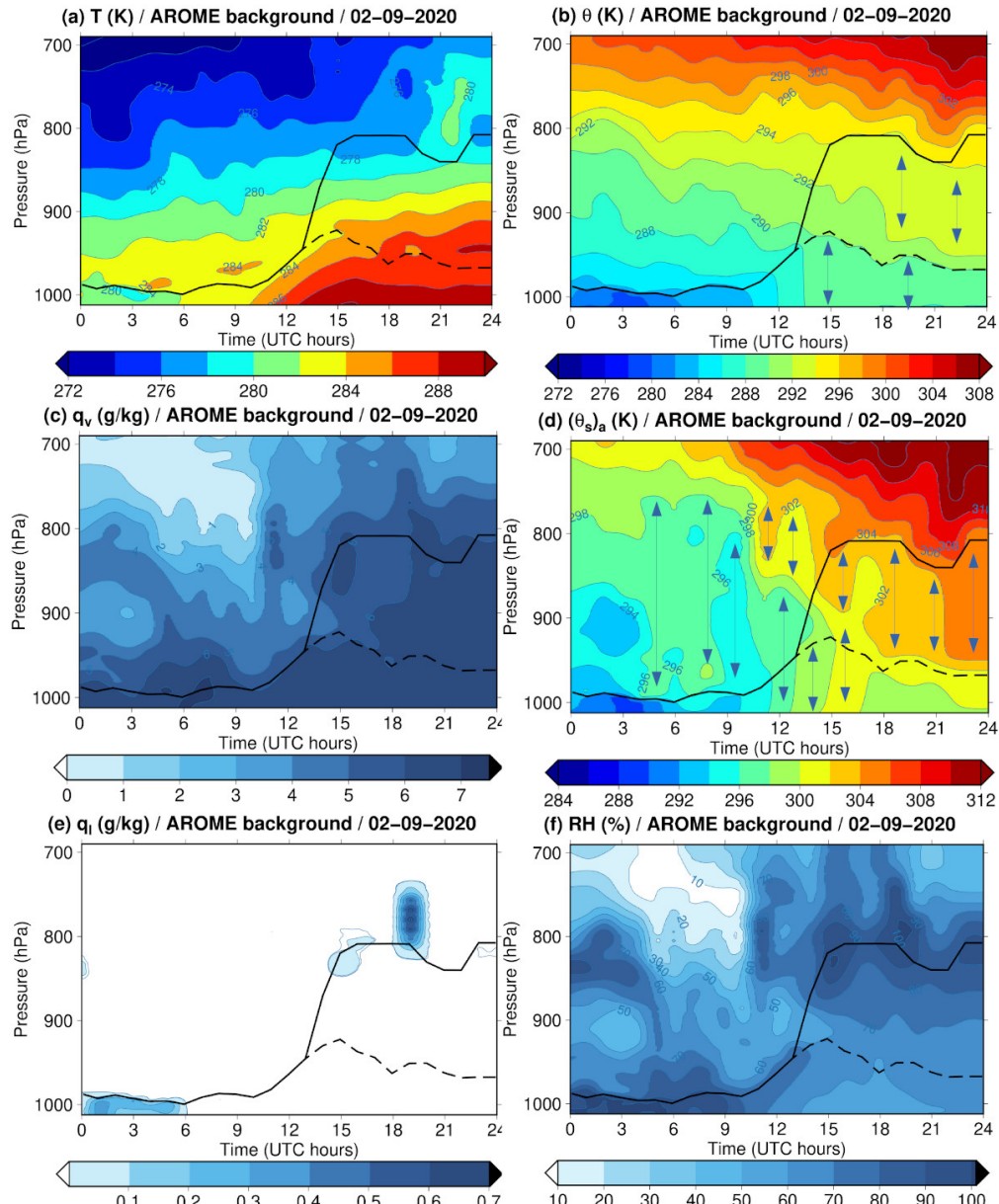

**Figure 2.** Vertical profiles derived from 1 h forecasts of AROME background for all hours of the day 9 February 2020 at Saint-Symphorien (Les Landes region in France) for: (a) absolute temperature $T$ every 2 K; (b) dry-air potential temperature $\theta$ every 0.2 K, (c) water-vapour specific content $q_v$ every 1 g/kg, (d) entropy potential temperature $(\theta_s)_a$ every 0.2 K, (e) cloud liquid-water specific content $q_l$ (contoured for 0.00001 g/kg and 0.002 g/kg, then every 0.1 g/kg above 0.1 g/kg) and (f) relative humidity (RH) every 10 %. The black curves (solid and dashed lines) represent the PBL heights determined from maximum of vertical gradients of $\theta$. The vertical arrows in (b) and (d) indicate areas where potential temperatures are almost homogeneous or constant along the vertical.

(at 23.84 GHz) was eliminated because of a receiver failure identified during the campaign. In this preliminary study, we have only considered the zenith observation geometry of the radiometer for the sake of simplicity.

The radiative transfer model $\mathcal{H}$ RTTOV-gb needed to simulate the model equivalent of the observations, together with the choice of the control vector and the specification of the background and observation error matrices, are presented in the next section.

## 3.2   The components of the 1D-Var

In 1D-Var systems, the integrated liquid water content, Liquid Water Path ($LWP$), can be included in the control vector $x$ as
initially proposed by Deblonde and English (2003) and more recently used by Martinet et al. (2020). A first experimental set-up has been defined where the minimization is performed with the control vector being $(T, q_v, LWP)$. It will be considered as the reference being named "REF". The 1D-Var system chosen for the present study is the one developed by the EUMETSAT NWP SAF[3], where the minimisation of the cost-function is solved using an iterative procedure proposed by Rodgers (1976) with a Gauss-Newton descent algorithm. During the minimization process, only the amount of integrated liquid water is changed. In
this approach, the two "moist" variables $q_v$ and $LWP$ are considered to be independent (no cross-covariances for background errors between these variables). The second experimental framework, where the control vector is $z = ((\theta_s)_a, q_t)$, corresponding to the conservative variables, is named "EXP". The numerical aspects of the 1D-Var minimisation are kept the same as in "REF".

Then, a set of reference matrices $\mathbf{B_x}(T, q_v)$ has been estimated every hour using the Ensemble Data Assimilation (EDA)
system of the AROME model on 9 February 2020. These matrices were obtained by computing statistics from a set of 25 members providing 3 h forecasts for a subset of 5000 points randomly selected in the AROME domain to obtain a sufficiently large statistical sample. Then, matrices associated with fog areas, and noted $\mathbf{B_x}(T, q_v)_{fog}$, were computed every hour by applying a fog mask (defined by areas where $q_l$ is above $10^{-6}$ $kg\,kg^{-1}$ for the three lowest model levels), in order to select only model grid points for which fog is forecast in the majority of the 25 AROME members. The background error covariance
matrices $\mathbf{B_z}((\theta_s)_a, q_t)$ and $\mathbf{B_z}((\theta_s)_a, q_t)_{fog}$ were obtained in a similar way.

The observation errors are those proposed by Martinet et al. (2020) with values between 1 and 1.7 K for humidity channels (frequencies between 22 and 31 GHz), values between 1 and 3 K for transparent channels affected by larger uncertainties in the modelling of the oxygen absorption band (frequencies between 51 and 54 GHz) and values below 0.5 K for the most opaque channels (frequencies between 55 and 58 GHz).

The RTTOV[4] radiative transfer model is used to calculate brightness temperatures in different frequency bands from atmospheric temperature, water vapour and hydrometeor profiles together with surface properties (provided by outputs from the AROME model). This radiative transfer model has been adapted to simulate ground-based microwave radiometer observations (RTTOV-gb) by De Angelis et al. (2016).

---

[3]Numerical Weather Prediction Satellite Application Facility

[4]Radiative Transfer for the TIROS Operational Vertical Sounder

## 4 Numerical results

The 1D-Var algorithm was tested on the day of 9 February 2020 with observations from the HATPRO microwave radiometer installed at Saint-Symphorien. This section presents and discusses the results obtained on three aspects: (1) the study of background error cross-correlations; (2) the performance of the 1D-Var assimilation system in observation space by examining the fit of simulated $TB$ with respect to the observed ones; and (3) in model space in terms of analysis increments for temperature, specific humidity and liquid water content.

### 4.1 The background error cross correlations

Figure 3 displays for the selected day at 06 UTC the cross-correlations between $T$ and $q_v$ (top) and between $(\theta_s)_a$ and $q_t$ (bottom), with (right) and without (left) fog mask. For the classical variables the correlations are strongly positive in the saturated boundary layer with the fog mask from levels 75 to 90 (between 1015 and 950 hPa), while with profiles in all-weather conditions the correlations between $T$ and $q_v$ are very weak in the lowest layers. On the other hand, the atmospheric layers above the fog layer exhibit negative correlations between temperature and specific humidity along the first diagonal.

When considering conservative variables, the correlations along the diagonal show a consistently positive signal in the troposphere (below level 20 located around 280 hPa). Contrary to the classical variables, which are rather independent in clear-sky atmospheres as previously shown by Ménétrier and Montmerle (2011), the $\mathbf{B_z}$ matrix reflects the physical link between the two new variables (shown by Eq. (4)) as diagnosed from the AROME model. The correlations are positive with and without fog mask. This result shows that the matrix $\mathbf{B_z}((\theta_s)_a, q_t)$ is less sensitive to fog conditions than the $\mathbf{B_x}$ matrix. It could therefore be possible to compute a $\mathbf{B_z}((\theta_s)_a, q_t)$ matrix without any profile selection criteria that would be nevertheless suitable for fog situations, resulting in a more robust estimate. This result is key for 1D-Var retrievals which are commonly used in the community of ground-based remote sensing instruments to provide databases of vertical profiles for the scientific community. In fact, the accuracy of 1D-Var retrievals is expected to be more robust with less flow-dependent $\mathbf{B}$ matrices.

It has also been noticed that these background error statistics are less dependent on the diurnal cycle and on the meteorological situation (e.g. in the presence of fog at 06 UTC and low clouds at 21 UTC), contrary to the $\mathbf{B_x}(T, q_v)$ matrix where there is a reduction in the area of positive correlation in the lowest layers between 06 UTC and 21 UTC (Fig. 4).

The 1D-Var results are now assessed in observation space by examining innovations (differences between observed and simulated brightness temperatures) from AROME background profiles and residuals. In the following, we have only used background error covariance matrices estimated at 06 UTC with fog mask, for a simplified comparison framework of the two 1D-Var systems.

### 4.2 1D-Var analysis fit to observations

Figure 5 presents both (a) innovations and (b,c) residuals obtained with the two 1D-Var systems (b: REF and c: EXP) for the 13 channels $(1, 2, 4 - 14)$ and for each hour of the day. The innovations are generally positive for water vapour sensitive channels during the day, and negative for temperature channels, especially in the morning. The differences are mostly between $-2.5$ and

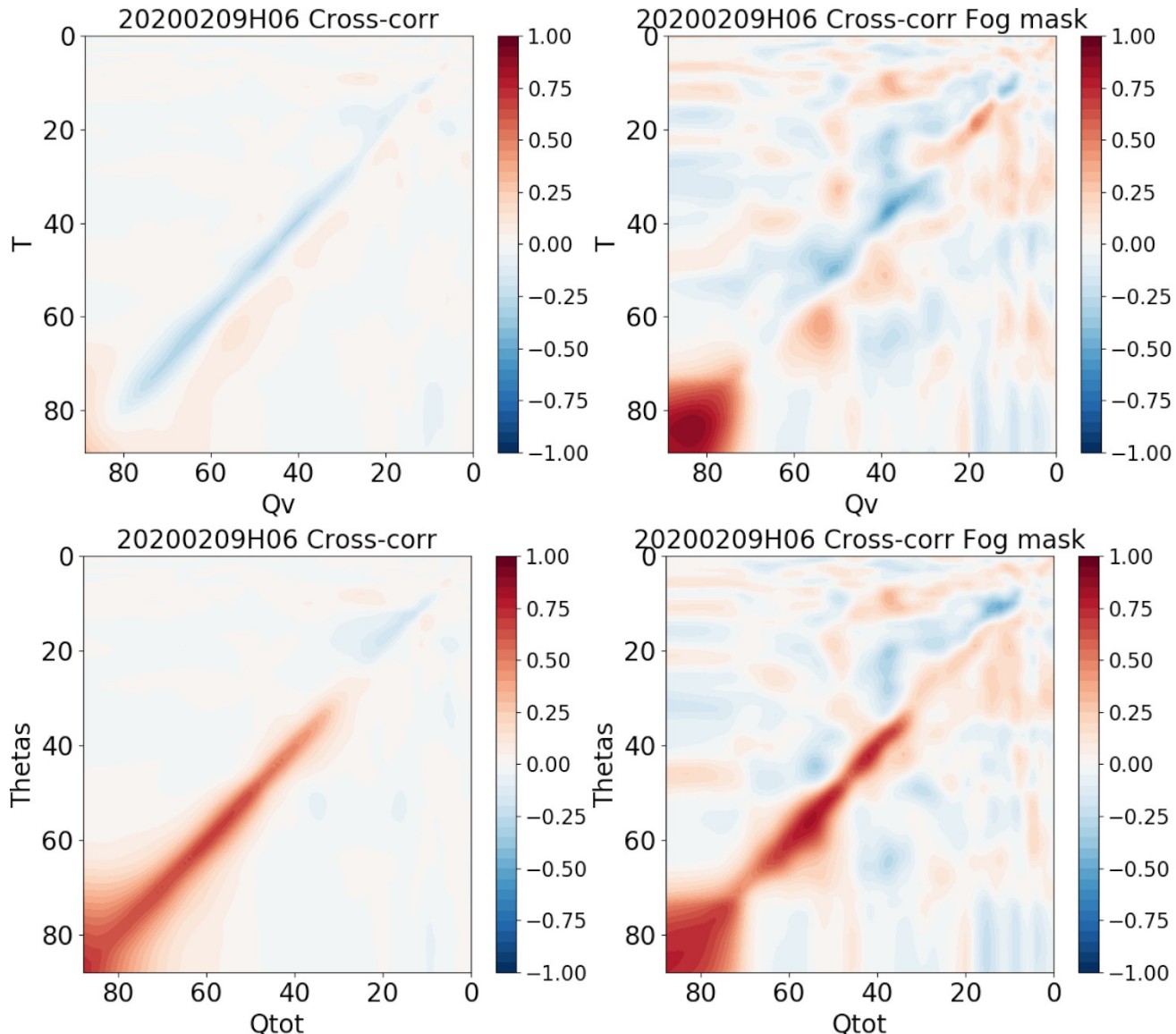

**Figure 3.** Background error cross-correlation matrices at 06 UTC 9 February 2020 with (right) and without (left) fog mask. Top: between the classical variables $(T, q_v)$ denoted in the axes by "T" and "Qv", respectively. Bottom: between the new conservative variables $((\theta_s)_a, q_t)$ denoted in the axes by "Thetas" and "Qtot", respectively. The axes correspond to the levels of the AROME vertical grid (1 at the top and 90 at the bottom). Correlations are between $-1$ and 1 units.

5 K. For channels 8, 9 and 10, which are sensitive to liquid water content, the innovations can reach higher values exceeding 10 K (in the afternoon) or being around $-5$ K (in the morning).

In terms of residuals, as expected from 1D-Var systems, both experiments significantly reduce the deviations of the observed $TB$ from those calculated using the background profiles, especially for the first eight channels sensitive to water vapour

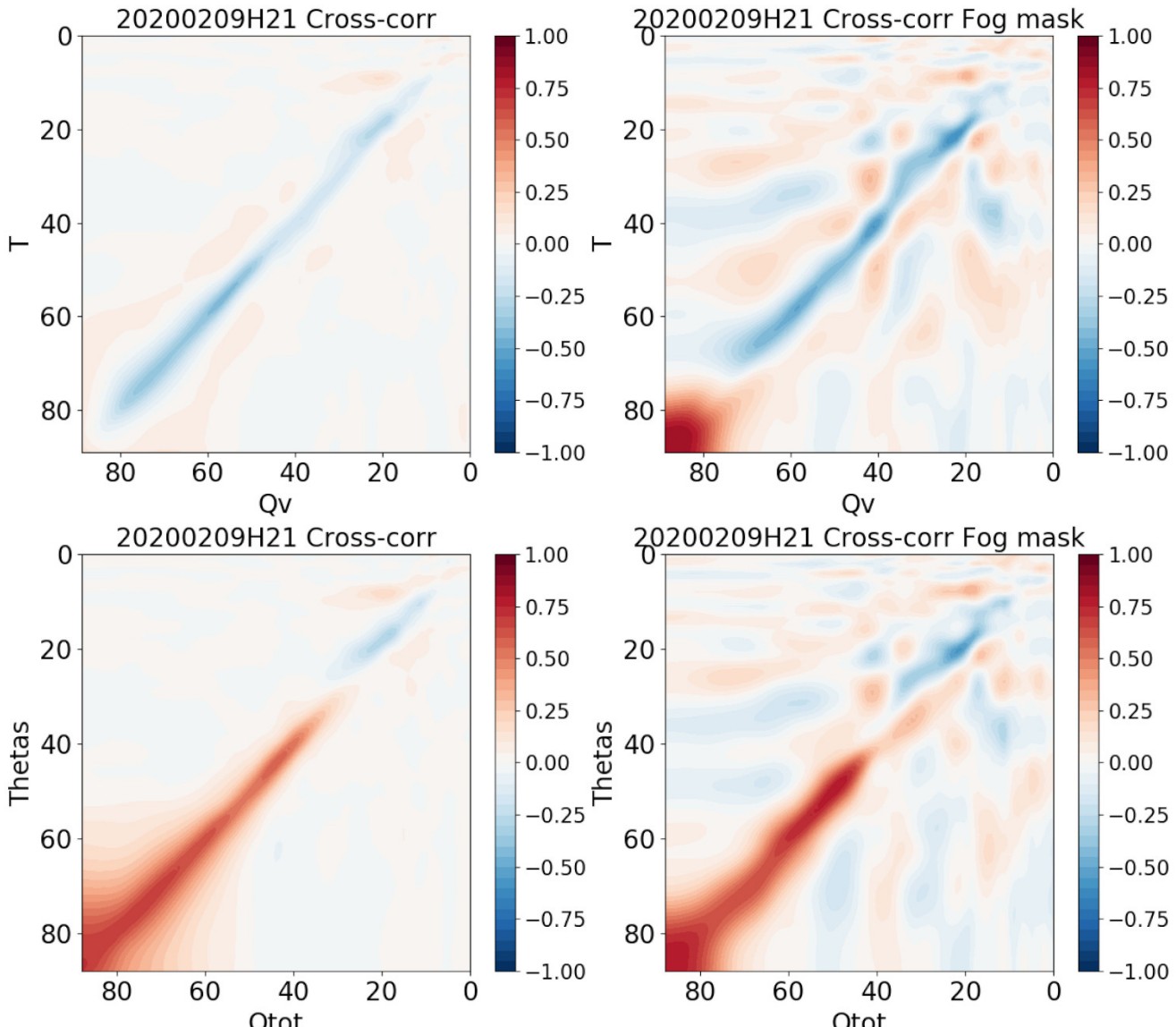

**Figure 4.** Same as Fig. 3, but at 21 UTC.

280    and liquid water. We can note that the residuals are not as reduced for channel 9 (52.28 GHz) compared to other channels. Indeed, channels 8 and 9 (51.26 and 52.28 GHz) suffer from larger calibration uncertainties (Maschwitz et al., 2013) and larger forward model uncertainties dominated by oxygen line mixing parameters (Cimini et al., 2018) than other temperature sensitive channels. However, by comparing simulated brightness temperatures (TB) with different absorption models (Hewison, 2007), or through a monitoring with simulated TB from clear-sky background profiles (Angelis et al., 2017; Martinet et al., 2020),

285    larger biases are generally observed only at 52.28 GHz. Consequently, the higher deviations observed in Fig. 5 for channel 9

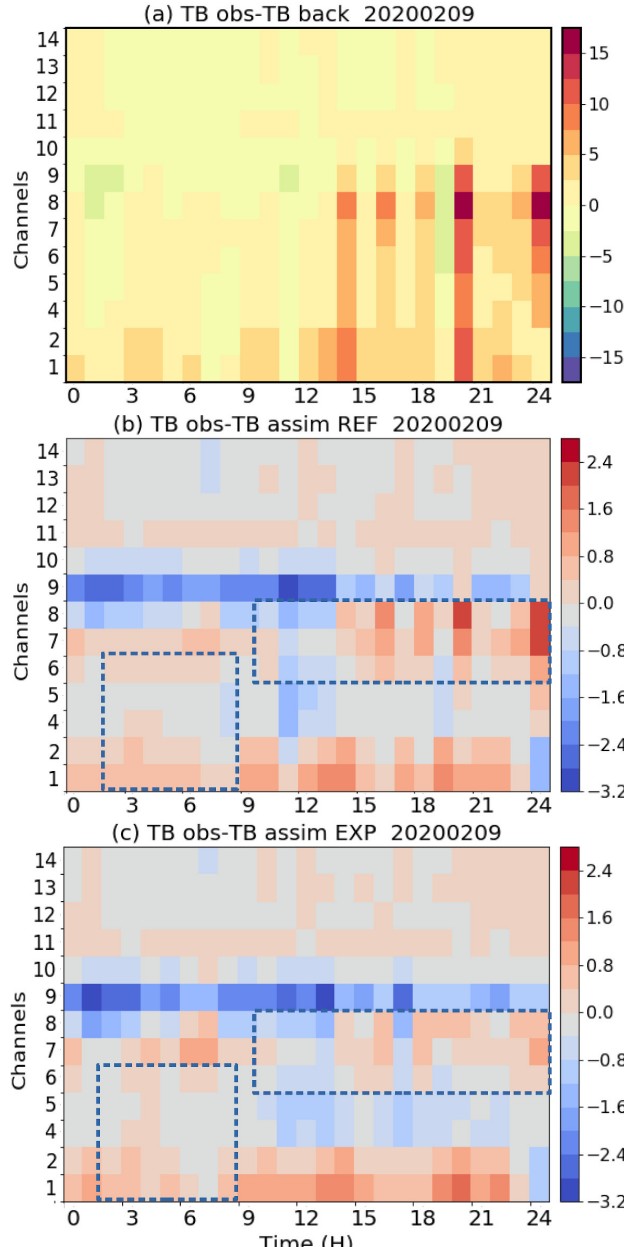

**Figure 5.** Differences in observed (channels 1, 2 and 4 to 7 being located between 22 and 31 GHz and channels 8 to 14 being located between 51 and 58 GHz, HATPRO radiometer) and simulated (with RTTOV-gb) brightness temperatures (in Kelvin): (a) from AROME background profiles; (b) from 1D-Var analyses from the REF configuration; and (c) from 1D-Var analyses from the EXP configuration for all hours of the day on 9 February 2020 at Saint-Symphorien (Les Landes region). The dashed blue boxes indicate the channels and hours where EXP is improved with respect to REF. Color-bars are in unit of K.

mostly originate from larger modelling and calibration uncertainties, which are taken into account in the assumed instrumental errors (prescribed observation errors of about 3 K for these two channels compared to < 1 K for other temperature sensitive channels) and also possibly from larger instrumental biases.

The temperature channels used in the zenith mode are modified less or very little, the deviations from the background values being much smaller than for the other channels. During the second half of the day, characterized by the presence of clouds around 800 hPa (see Figs. 2 (e) and (f)), the residual values are largely reduced in the frequency bands sensitive to liquid water for channels 6, 7 and 8, especially for EXP as shown by the comparison of the pixels in the dashed rectangular boxes in Figs. 5 (b) and (c). Residuals are also slightly reduced for EXP in the morning and during the fog and low temperature period for the first five channels $(1, 2, 4-6)$ between 2 and 8 UTC.

**Table 2.** Bias / RMSE (K) of the background and analyses produced by EXP and REF against MWR TB observations. Statistics are computed either using all data or restricting to channels number 1 to 5 between 2 and 8 UTC or channels 7 to 9 between 10 and 24 UTC (these two sub-samplings represent the dashed rectangular boxes in Figs. 5 (b) and (c)).

|  | Background | REF | EXP |
|---|---|---|---|
| All data | 1.3 / 2.2 | -0.11 / 0.72 | -0.17 / 0.71 |
| Channels 1 to 6, 2 to 8 UTC | 1.5 / 2.2 | 0.11 / 0.3 | 0.08 / 0.3 |
| Channels 6 to 8, 10 to 24 UTC | 2.7 / 4.3 | 0.16 / 0.57 | -0.12 / 0.37 |

In order to quantify these results on 9 February 2020 dataset (all hours and all channels), the bias and root mean square ($RMS$) error values are computed for the background and the analyses produced by REF and EXP. The innovations are characterized by a $RMS$ error of 3.20 K and a bias of 1.32 K. Both assimilation experiments reduce these two quantities by modifying model profiles. The $RMS$ errors are 0.71 K for EXP and 0.72 K for REF and the biases are $-0.17$ K for EXP and $-0.11$ K for REF. These statistics have also been calculated by restricting the dataset to the two dashed rectangular boxes represented in Figs. 5 (b) and (c). A significant improvement is observed for the most sensitive channels to liquid water in the afternoon with a RMSE decreased from 4.3 K in the background to 0.57 K in REF and 0.37 K in EXP. For all computed statistics, EXP always provides the best performance in terms of RMSE. Table 2 summarizes the bias and RMSE values obtained for the different samples.

### 4.3 Vertical profiles of analysis increments

After examining the fit of the two experiments to the observed $TB$s, we assess the corrections made in model space. Figure 6 shows the increments of (a), (b) temperature, (c), (d) specific humidity, and (e), (f) liquid water for the two experiments REF (left panels) and EXP (right panels). In addition, the increments of $(\theta_s)_a$ are shown in (g)-(h).

The temperature increments are mostly located in the lower troposphere (below 650 hPa) with a dominance of negative values of small amplitude (around 0.5 K). This is consistent with negative innovations observed on temperature channels highlighting a warm bias in the background profiles. The areas of maximum cooling take place in cloud layers (inside the thick

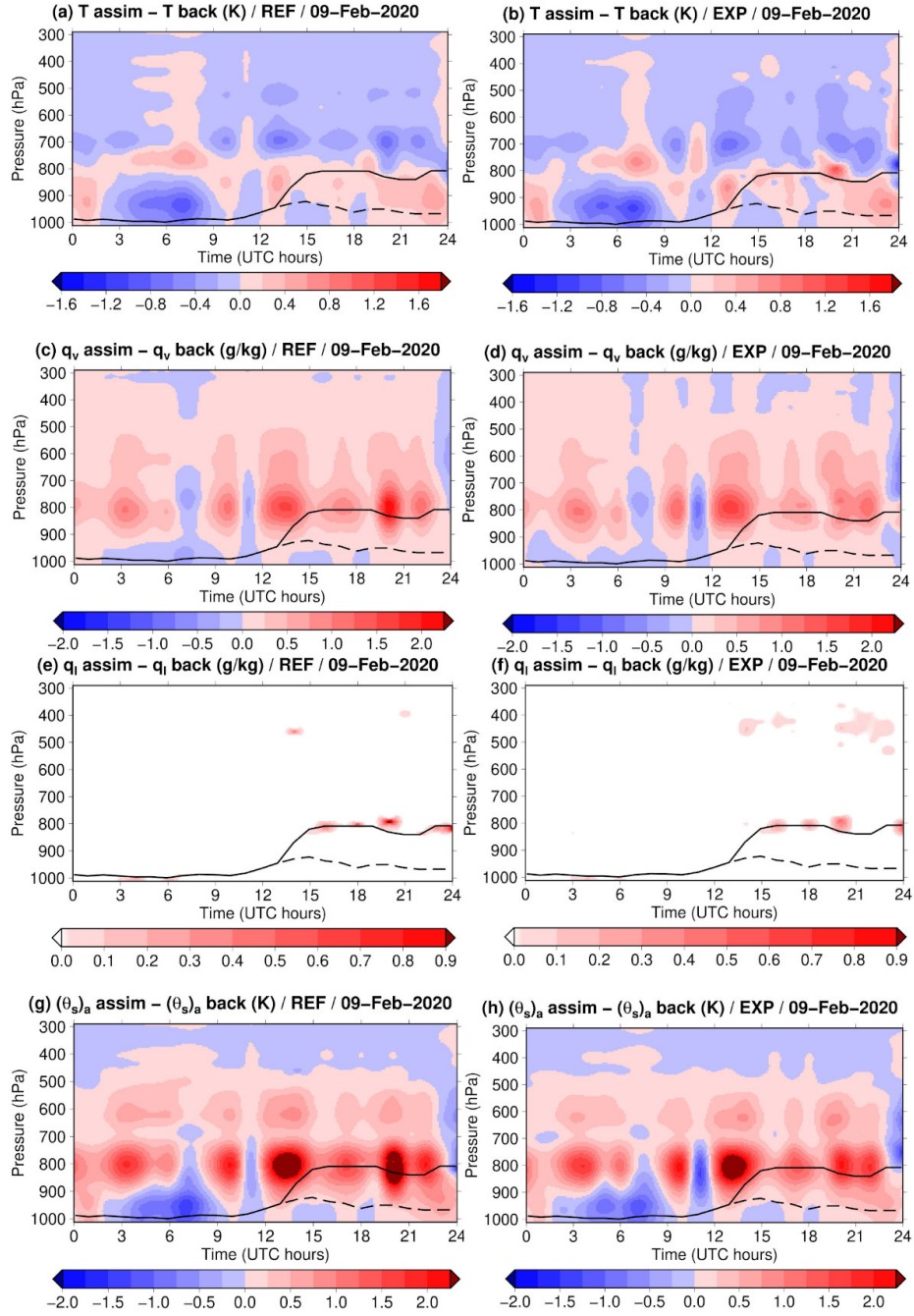

**Figure 6.** Profiles of analysis increments resulting from two 1D-Var experiments: REF (left) and EXP (right), for: (a)-(b) $T$ in K; (c)-(d) $q_v$ in $\mathrm{g\,kg^{-1}}$; (e)-(f) $q_l$ in $\mathrm{g\,kg^{-1}}$ and (g)-(h) $(\theta_s)_a$ in K.Color-bars have the same units ( K or $\mathrm{g\,kg^{-1}}$ ) as the variables.

fog layer below 900 hPa until 9 UTC and around 700 hPa after 12 UTC). The increments are rather similar between REF and EXP, but the positive increments appear to be larger with EXP (e.g. at 08 and 20 UTC around 800 hPa).

Concerning the profiles associated with moist variables, the structures show similarities between the two experiments but with differences in intensity. During the night and in the morning, the $q_v$ increments near the surface are negative. These negative increments are projected into increments having the same sign as $T$ by the strong positive cross-correlations of the $\mathbf{B}_{fog}$ matrix up to 900 hPa (Fig. 3). Thus, the largest negative temperature and specific humidity increments remain confined in the lowest layers.

Liquid water is added in both experiments between 03 UTC and 07 UTC, close to the surface, where the Jacobians of the most sensitive channels to $q_l$ (6 to 8) have significant values in the fog layer present in the background (see Fig. 2e). After 14 UTC, values of $q_v$ between 850 and 700 hPa and $q_l$ around 800 hPa are enhanced in both cases, with larger increments for the REF case, in particular at 20 UTC and around midnight. Most of the liquid water is created in low clouds. Additionally, increments of $q_l$ above 600 hPa are larger and more extended vertically and in time in EXP, where condensation occurs over a thicker atmospheric layer between 500 hPa and 300 hPa after 12 UTC. In the REF experiment, the creation of liquid water above 500 hPa only reaches values of 0.3 g/kg sporadically, for example at 21 UTC. In this experimental set-up, condensed water can be created or removed over the whole column by means of the supersaturation diagnosed at each iteration of the minimisation process (since RTTOV-gb needs $(T, q_v, q_l)$ profiles for the $TB$ computation.) This is a clear advantage of EXP over REF, which keeps the vertical structure of the $q_l$ profile unchanged from the background. In REF, liquid water is only added where it already exists in the background because once the $LWP$ variable is updated, the analyzed $q_l$ profile is just modified proportionally to the ratio between the $LWP$ of the analysis and of the background, as explained in more details by Deblonde and English (2003).

The profiles of increments for $(\theta_s)_a$ show structures similar to the increments of $q_v$ around 800 hPa and to the increments of $T$ below, where temperature Jacobians are the largest (see Fig. 7 from De Angelis et al. (2016)). The conversion of $T$, $q_v$ and $q_t$ changes obtained with REF into $(\theta_s)_a$ increments (Fig. 6g) highlights the main differences between the two systems. They take place around 800 hPa with larger increments produced by the new 1D-Var particularly between 11 and 14 UTC.

Some radiosoundings (RS) have been launched during the SOFOG3D IOPs. As only one RS profile was launched at $05:21$ UTC on the case study presented in the article, no statistical evaluation of the profile increments can be carried out. However, we have conducted an evaluation of the analysis increments obtained at 5 and 6 UTC (the 1D-Var retrievals being performed at a 1h temporal resolution in line with the operational AROME assimilation cycles) around the RS launched time. As the AROME temperature background profile extracted at 6 UTC was found to have a vertical structure closer to the RS launched at $05:21$ UTC, figures 7 compare the AROME background profile and 1D-Var analyses performed with the REF and EXP experiments valid at 6 UTC against the RS profile.

The temperature increments are a step in the right direction by cooling the AROME background profile in line with the observed RS profile. The two 1D-Var analyses are close to each other, but the EXP analysis produces a temperature profile slightly cooler compared to the REF analysis. In terms of absolute humidity ($\rho_v = p_v/(R_v T)$, with $p_v$ the partial pressure and $R_v$ the gas constant for water vapor), the background profile already exhibits a similar structure compared to the RS profile.

The 1D-Var increments are thus small and close between the two experiments. However, we can note that the EXP profile is slightly moister than the REF profile from the surface up to 3500 m which leads to a somewhat better agreement with the RS profile below 1500 m. In terms of integrated water vapor (IWV), a significant improvement of the background IWV with respect to the RS IWV is observed with a difference reduced from almost $1 \ \mathrm{kg \, m^{-2}}$ in the background to less than $0.4 \ \mathrm{kg \, m^{-2}}$ in the analyses. These analyses confirm the improvement brought to the model profiles by both the REF and EXP analysis increments, with some enhanced improvement for EXP.

## 5  Conclusions

The aim of this study was to examine the value of using moist-air entropy potential temperature $(\theta_s)_a$ and total water content $q_t$ as new control variables for variational assimilation schemes. In fact, the use of control variables less dependent to vertical gradients of $(T, q_v, q_l, q_i)$ variables should ease the specification of background error covariance matrices, which play a key role in the quality of the analysis state in operational assimilation schemes.

To that end, a 1D-Var system has been used to assimilate brightness temperature $(TB)$ observations from the ground-based HATPRO microwave radiometer installed at Saint-Symphorien (Les Landes region in South-Western France) during the SOFOG3D measurement campaign (winter 2019-2020).

The 1D-Var system has been adapted to consider these new quantities as control variables. Since the radiative transfer model needs profiles of temperature, water vapour and cloud liquid water for the simulation of $TB$, an adjustment process has been defined to obtain these quantities from $(\theta_s)_a$ and $q_t$. The adjoint version of this conversion has been developed for an efficient estimation of the gradient of the cost-function. Dedicated background error covariance matrices have been estimated from the Ensemble Data Assimilation system of AROME. We first demonstrated that the matrices for the new variables are less dependent on the meteorological situation (all-weather conditions vs. fog conditions) and on the time of the day (stable conditions by night vs. unstable conditions during the day) leading to potentially more robust estimates. This is an important result as the optimal estimation of the analysis depends on the accurate specification of the background error covariance matrix which is known to highly vary with weather conditions when using classical control variables.

The new 1D-Var has produced rather similar results in terms of fit of the analysis to observed $TB$ values when compared to the classical one using temperature, water vapour and liquid water path. Nevertheless, quantitative results reveal smaller biases and $RMS$ values with the new system in low cloud and fog areas. It has also been noticed that atmospheric increments are somewhat different in cloudy conditions between the two systems. For example, in the stratocumulus layer that formed during the afternoon, the new 1D-Var induces larger temperature increments and reduced liquid water corrections. Moreover, its capacity to generate cloud condensates in clear-sky regions of the background has been demonstrated. As a preliminary validation, the retrieved profiles from the 1D-Var have been compared favourably against an independent observation data set (one radiosounding launched during the SOFOG3D field campaign). The new 1D-Var leads to profiles of temperature and absolute humidity slightly closer to observations in the planetary boundary layer.

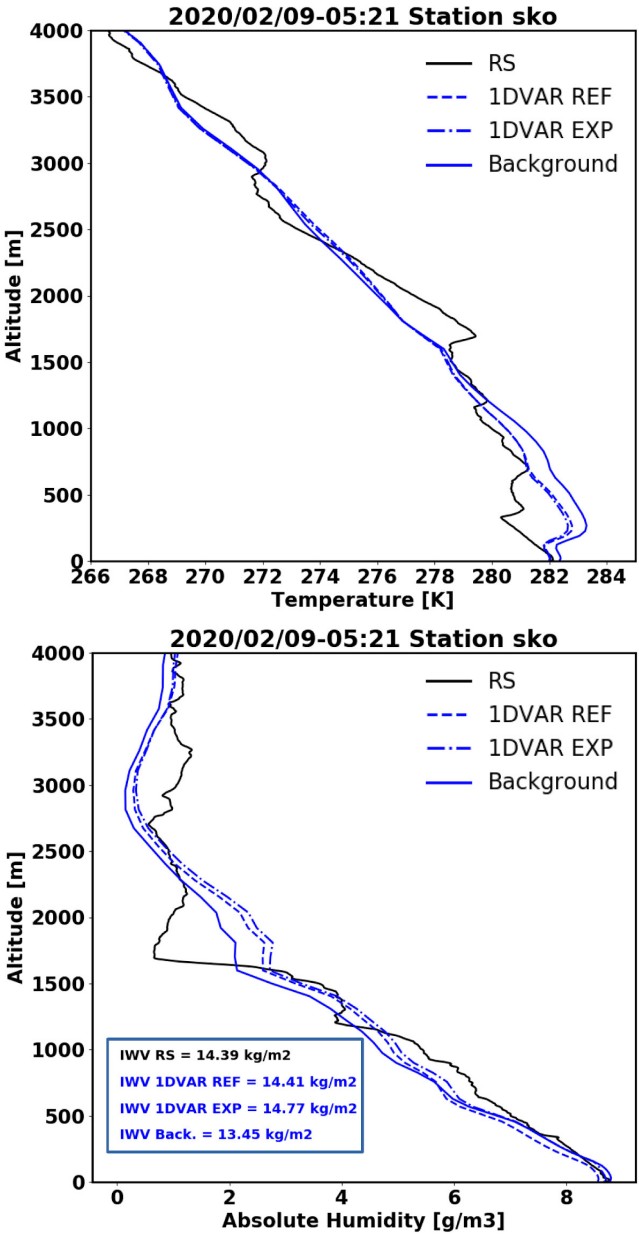

**Figure 7.** Vertical profiles of absolute temperature $T$ (top panel, in K) and absolute humidity $\rho_v$ (bottom panel, in $\mathrm{g\,m^{-3}}$) for the 9 February 2020 and showing: the RS launched at $05:21$ UTC (solid black); the AROME background valid at 06 UTC (solid blue); and the 1D-Var retrievals at 06 UTC obtained with REF (dashed blue) and EXP (dotted-dashed blue). Integrated water vapor (IWV) retrievals are also compared to the RS IWV (in the blue box in the bottom panel).

These encouraging results obtained from this feasibility study need to be consolidated by complementary studies. Observed brightness temperatures at lower elevation angles should be included in the 1D-Var for a better constraint on temperature profiles within the atmospheric boundary layer. Indeed, larger differences in the temperature increments might be obtained between the classical 1D-Var system and the 1D-Var system using the new conservative variables when additional elevation angles are included in the observation vector. Other case studies from the field campaign could also be examined to confirm our first conclusions.

Finally, the conversion operator could be improved by accounting not only for liquid water content $q_l$ but also for ice water content $q_i$ (e.g. using a temperature threshold criteria). Indeed, inclusion of $q_i$ in the conversion operator should lead to more realistic retrieved profiles of cloud condensates, and a 1D-var system with only $q_l$ can create water clouds at locations where ice clouds should be present, as done in our experiment around 400 hPa between 15 and 24 UTC. However, since the frequencies of HATPRO are not sensitive to ice water content, the fit of simulated $TB$s to observations could be reduced. In consequence, the synergy with an instrument sensitive to ice water clouds, such as a W-band cloud radar, would be necessary for improved retrievals of both $q_i$ and $q_l$. It is worth noticing that the variable $(\theta_s)_a$ can easily be generalized to the case of the ice phase and mixed phases by taking advantage of the general definition of $\theta_s$ and $(\theta_s)_1$, where $L_{vap}q_l$ is simply replaced by $L_{vap}q_l + L_{sub}q_i$.

*Code and data availability.* The numerical code of the radiative transfer model RTTOV-gb together with the associated resources (coefficient files) can be downloaded from http://cetemps.aquila.infn.it/rttovgb/rttovgb.html and from https://nwp-saf.eumetsat.int/site/software/rttov-gb/. The 1D-Var software has been adapted from the NWP SAF 1D-Var provided here: https://nwp-saf.eumetsat.int/site/software/1d-var/ and is available on request to pauline.martinet@meteo.fr. The instrumental data are available on the AERIS website dedicated to the SOFOG3D field experiment: https://sofog3d.aeris-data.fr/catalogue/. AROME backgrounds are available on request to pauline.martinet@meteo.fr. Quicklooks from the cloud radar BASTA are available on: http://basta.projet.latmos.ipsl.fr. The BUMP library to compute background error matrices, developed in the framework of the JEDI project led by the JCSDA (Joint Center for Satellite Data Assimilation, Boulder, Colorado), can be downloaded at https://github.com/JCSDA/saber.

*Author contributions.* PMarq supervised the work of AB, contributed to the implementation of the new conservative variables in the computation of new background error covariance matrices, participated to the scientific analysis and manuscript redaction. JFM developed the conversion operator and adjoint version, participated in the scientific analysis and manuscript redaction. PMart supervised the modification of the 1D-Var algorithm, supported the use of the EDA to compute background error covariance matrices, provided the instrumental data used in the 1D-Var and participated in the manuscript revision. AB adapted the 1D-Var algorithm and processed all the data, prepared the figures and participated in the manuscript revision. BM developed and adapted the BUMP library to compute the background error covariance matrices for the 1D-Var, and participated in the manuscript revision.

*Competing interests.* The authors declare that they have no conflict of interest.

*Acknowledgements.* The authors are very grateful to the two anonymous reviewers who suggested substantial improvements to the article. The instrumental data used in this study are part of the SOFOG3D experiment. The SOFOG3D field campaign was supported by METEO-FRANCE and ANR through grant AAPG 2018-CE01-0004. Data are managed by the French national center for Atmospheric data and services AERIS. The MWR network deployment was carried out thanks to support by IfU GmbH, the Köln University, the Met-Office, Laboratoire d'Aérologie, Meteoswiss, ONERA, and Radiometer Physics GmbH. MWR data have been made available, quality controlled and processed in the frame of CPEX-LAB (Cloud and Precipitation Exploration LABoratory, www.cpex-lab.de), a competence center within the Geoverbund ABC/J by acting support of Ulrich Löhnert, Rainer Haseneder-Lind and Arthur Kremer from the University of Cologne. This collaboration is driven by the European COST actions ES1303 TOPROF and CA18235 PROBE. Julien Delanoë and Susana Jorquera are thanked for providing the cloud radar quicklooks used in this study for better understanding the meteorological situation. Thibaut Montmerle and Yann Michel are thanked for their support on the use of the AROME EDA to compute background error covariance matrices. The work of Benjamin Ménétrier is funded by the JCSDA (Joint Center for Satellite Data Assimilation, Boulder, Colorado) UCAR SUBAWD2285.

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
