# Peer review of "Towards the use of conservative thermodynamic variables in data assimilation: a case study using ground-based microwave radiometer measurements"

_Atmospheric Measurement Techniques, 2021_

## Referee Comment (RC1)

**Review of amt-2021-361: Towards the use of conservative thermodynamic variables in data assimilation: preliminary results using ground-based microwave radiometer measurements, Marquet et al.**

The paper demonstrates the analyses of model profiles with the assimilation of groundbased microwave radiometer (MWR) brightness temperatures with a 1D-Var scheme and the use of two conservative thermodynamic variables as control variables in a case study. The presented work shows the potential of using the described conservative thermodynamic variables for 1D-Var MWR retrievals, and the benefit of the obtained, less weather-dependent, model error covariance matrices. It also hints at the benefit of MWR data for data assimilation in the studied fog case and is therefore a valuable step towards the use of MWR data in operational NWP systems.

The methodology is well explained, the experiment is carefully analysed and results are clearly presented. The assessment is done both in observation and model space. The manuscript is well written, and the figures are illustrative of the relevant results.

I support the publication of this article after the authors addressed the comments and suggestions listed below.

**General comments**

- **2.1 The moist-air entropy potential temperature:** The section explains the most important points in the derivation of the moist-air entropy potential temperature and provides an extensive list of references for further information. Although it is beyond the scope of this manuscript to go in full details here, I found the description of the conservation laws (I. 74ff) a little short and difficult to follow for people not familiar with these variables. Some sentences are also quite long and complicated. I would very much welcome a reformulation of this paragraph.
- **4.3 Vertical profiles of analysis increments and 5 Conclusions:** Do you have the possibility to make a stronger link between the increments and potential improvements in the model profiles? Are the increments located at the right place and going in the desired direction? You provide a good indication for temperature increments with respect to MWR measurements in I. 271ff. A comparison to soundings, as you mention in I. 321, could be especially valuable. Would it be possible to provide a preliminary assessment?

**Specific comments**

- **Title:** I suggest changing the title to 'Towards the use of conservative thermodynamic variables in data assimilation: a case study using ground-based microwave radiometer measurements' to make the scope of the manuscript clearer.
- 2.2 The 1D-Var formalism: Very clear and concise description of 1D-Var, pleasant to read! It would maybe be good to include a reference to a more detailed description of 1D-Var theory.
- 2.3 The conversion operator: To help make it easier to understand, it would be helpful to add in this section that this conversion from  $((\theta_s)_a, q_t)$  to  $(T, q_v, q_l)$  is needed to feed RTTOV after the analysis.

- 4.1 The background error cross correlations: I am not familiar with the variable (θs)a. Could you make a little clearer why you expect a link between (θs)a and qt in clear-sky atmosphere (I. 234)? If this should be evident from section 2.1 it would maybe be good to highlight it more.
- **4.2 1D-Var analysis fit to observations:** In order to better compare the REF and EXP analyses in the regions of concern, it might be interesting to compute and compare RMSE and bias only within the blue rectangular boxes.
- 5 Conclusions:
  - **I. 300ff** Maybe this paragraph would better suit to section 2.1, where the variables are introduced and described.
  - **I. 326ff** Do you expect results of TB to vary a lot after including  $q_i$  in the conversion operator? Where there ice clouds in the presented case (1. 327)?

**Technical corrections**

**Text:**

| I. 5:     | '[] that are currently highly dependent on weather conditions []' $\rightarrow$ '[] that are highly dependent on weather conditions when using classical variables, []' |
|-----------|-------------------------------------------------------------------------------------------------------------------------------------------------------------------------|
| I. 28:    | 'scheme' $\rightarrow$ 'schemes'                                                                                                                                        |
| I. 38:    | '[] that data assimilation systems used to be []' $\to$ '[] that most data assimilation systems are []'                                                                 |
| I. 67:    | Use $s_{d0}(T_0, p_0)$ in the formula, as in I. 69.                                                                                                                     |
| I. 71:    | State that the explanation for $\Lambda_r$ follows later in the section.                                                                                                |
| I. 73:    | '[] sublimation.' $ ightarrow$ '[] sublimation, respectively.'                                                                                                          |
| I. 106:   | Remove the point at the end of the title.                                                                                                                               |
| I. 143:   | 'set-up' $\rightarrow$ 'setup'                                                                                                                                          |
| I. 147:   | '[] such a HATPRO []' $ ightarrow$ '[] such as a HATPRO []'                                                                                                             |
| I. 155:   | 'lowest' instead of 'first few'                                                                                                                                         |
| I. 192:   | I propose to mention RTTOV-gb here after ${\cal H}$ because it was already mentioned earlier in the text.                                                               |
| I. 195:   | comma after '(LWP)'                                                                                                                                                     |
| I. 201:   | No new paragraph before I. 201                                                                                                                                          |
| I. 236-23 | 57: $\mathbf{B}_{\mathbf{z}}(\theta_s, q_t) \to \mathbf{B}_{\mathbf{z}}((\theta_s)_a, q_t)$ ?                                                                    |
| I. 276:   | 'similarity' $ ightarrow$ 'similarities'                                                                                                                                |

- **I. 284-287:** Suggestion: 'Most of the liquid water is created in low clouds. Additionally, increments of  $q_l$  above 600 hPa are larger and more extended vertically and in time in EXP, where condensation occurs over a thicker atmospheric layer between 500 hPa and 300 hPa after 12 UTC. In the REF experiment, the creation of liquid water above 500 hPa only reaches values of 0.3 g/kg sporadically, for example at 21UTC. In the EXP setup, [...]'
- **1. 289:** '[...] keeps unchanged the vertical structure of the  $q_l$  profile [...]'  $\rightarrow$  '[...] keeps the vertical structure of the  $q_l$  profile unchanged [...]'
- **I. 289-290:** 'Liquid water is added where it already existed  $[...]' \rightarrow$  'In REF, liquid water is only added where it already exists [...]'
- **I. 293:** '[...] show similar structures to [...]'  $\rightarrow$  '[...] show structures similar to [...]'
- **I. 296:** 'value' instead of 'interest'?
- **I. 297:** 'for assimilating'  $\rightarrow$  'to assimilate'
- I. 299: 'over South-Western France'  $\rightarrow$  'in South-Western France'
- I. 303: '(in e.g. fronts, [....])'
- **I. 305:** 'significance' instead of 'interest'?
- **I. 324:** 'increment'  $\rightarrow$  'increments'

**Figures:**

- Figure 1: Would it be possible to provide this figure with slightly larger font size?
- Figure 2:
  - Do you mean 2K instead of 0.2K in the caption?
  - 'Relative Humidity' without capital letters
  - Could you add a description of the arrows (2(b), 2(d)) in the caption?
- Figure 5:
  - Please make the numbering of the channels consistent between figure (1-13) and caption (0-12). Would it maybe be possible to even enumerate the channels in the figure after the MWR channel numbers (1-2,4-14 as channel 3 was removed, if I am correct)?
  - Provide units to the color bar.
  - I find it a little difficult to compare the values between panels (a) and (b)/(c) and really point out the magnitude of the reduction of deviations. Would it be possible to use the same range in the color bar of all three panels? Or alternatively at least point out the different scaling in the caption for clarity.
- Figure 6: I suggest centering the color map to fit 0 to the change of color between blue and red.

---

## Author Comment (AC1)

**Answers to the Referee 1 about the paper amt-2021-361:**

"Towards the use of conservative thermodynamic variables in data assimilation: **preliminary results a case study** using ground-based microwave radiometer measurements"

> by Pascal Marquet, Pauline Martinet, Jean-François Mahfouf, Alina Lavinia Barbu, and Benjamin Ménétrier.

> > January 21, 2022

**1) Answers to the referee RC1**

• The paper demonstrates the analyses of model profiles with the assimilation of ground- based microwave radiometer (MWR) brightness temperatures with a 1D-Var scheme and the use of two conservative thermodynamic variables as control variables in a case study. The presented work shows the potential of using the described conservative ther- modynamic variables for 1D-Var MWR retrievals, and the benefit of the obtained, less weather-dependent, model error covariance matrices. It also hints at the benefit of MWR data for data assimilation in the studied fog case and is therefore a valuable step towards the use of MWR data in operational NWP systems.

• The methodology is well explained, the experiment is carefully analysed and results are clearly presented. The assessment is done both in observation and model space. The manuscript is well written, and the figures are illustrative of the relevant results.

• I support the publication of this article after the authors addressed the comments and suggestions listed below.

We thank you for this review and have taken into account almost all your comments and suggestions, as described below.

Please find, also, a document in "difference" mode where it is possible to isolate all changes in colour mode (from red to blue).

**2) General comments**

**Section 2.1 / The moist-air entropy potential temperature**

• The section explains the most important points in the derivation of the moist-air entropy potential temperature and provides an extensive list of references for further information. Although it is beyond the scope of this manuscript to go in full details here, I found the description of the conservation laws (line 74ff) a little short and difficult to follow for people not familiar with these variables. Some sentences are also quite long and complicated. I would very much welcome a reformulation of this paragraph.

Several of the sentences have been rewritten in this paragraph, and some of the sentences in the conclusion have been moved in this paragraph.

**Section 4.3 (Vertical profiles of analysis increments) and Section 5 (Conclusions)**

• Do you have the possibility to make a stronger link between the increments and potential improvements in the model profiles?

Two new figures 7 have been introduced and commented in the text at the end of section 4.3.

It is explained that:

"Some radiosoundings (RS) have been launched during the SOFOG3D IOPs. As only one RS profile was launched at 05:21~UTC on the case study presented in the article, no statistical evaluation of the profile increments can be carried out. However, we have conducted an evaluation of the analysis increments obtained at 5 and 6 UTC (the 1D-Var retrievals being performed at a 1h temporal resolution in line with the operational AROME assimilation cycles) around the RS launched time. The AROME temperature background profile extracted at 6 UTC has a vertical structure close to the RS launched at 05:21~UTC.

Figs 7 compare the AROME background profile and 1D-Var analyses performed with the REF and EXP experiments against the RS profile. The temperature increments are a step in the right direction by cooling the AROME background profile in line with the observed RS profile. The two 1D-Var analyses are close to each other, but the EXP analysis produces a temperature profile slightly cooler compared to the REF analysis. In terms of absolute humidity ( $\rho_v = p_v/(R_v T)$ , with  $p_v$  the partial pressure and  $R_v$  the gas constant for water vapour), the background profile already exhibits a similar structure compared to the RS profile. The EXP profile is slightly moister than the REF profile from the surface up to 3500 m which leads to a somewhat better agreement with the RS profile below 1500 m. The 1D-Var increments are thus small and close between the two experiments, although an improvement of the integrated water vapor (IWV) is observed when compared to the background, with a difference against the RS IWV reduced from almost 1 kg m-2 in the background to less than 0.4 kg m-2 in the 1D-Var analyses. These analyses confirm the improvement brought to the model profiles by both the REF and EXP analysis increments, with some enhanced improvement for EXP. "

• Are the increments located at the right place and going in the desired direction?

One can notice from results displayed in Figure 7 that, in the boundary layer (below 1000 m), the corrections induced by the assimilation are improving the profiles with respect to radiosounding data (temperature decrease and humidity increase). The analysis profiles from EXP are slightly closer to the RS profiles than those from REF.

• You provide a good indication for temperature increments with respect to MWR measurements in line 271ff. A comparison to soundings, as you mention in line 321, could be especially valuable. Would it be possible to provide a preliminary assessment?

This preliminary assessment is shown by the addition of Figure 7 in the revised paper and the corresponding discussion given section 4.3.

**3) Specific comments**

**Title**

• I suggest changing the title to 'Towards the use of conservative thermodynamic variables in data assimilation: a case study using ground-based microwave radiometer measurements' to make the scope of the manuscript clearer.

The title has been modified as suggested.

**Section 2.2**

• The 1D-Var formalism: Very clear and concise description of 1D-Var, pleasant to read! It would maybe be good to include a reference to a more detailed description of 1D-Var theory.

Two additional references are given in this section regarding 1D-Var retrieval theory: Rodgers (1976) and Thépaut and Moll (1990).

**Section 2.3**

• The conversion operator: To help make it easier to understand, it would be helpful to add in this section that this conversion from  $((\theta_s)_a, q_t)$  to  $(T, q_v, q_l)$  is needed to feed RTTOV after the analysis.

Additional explanation is given regarding the use of the conversion operator in the 1D-Var: "The second term on the right hand side of Eq. (11) and (12) indicates that the conversion operator  $\mathcal{L}$  is needed to compute the brightness temperatures from the observation operator  $\mathcal{H}$ . Indeed RTTOV-gb requires profiles of temperature, specific humidity and liquid water content as input quantities. This space change is required at each step of the minimisation process. For the computation of the gradient of the costfunction  $\nabla_z \mathcal{J}$ , the linearized version (adjoint) of  $\mathcal{L}$  is also necessary. In practice, the operator  $\mathbf{L}^T$ provides the gradient of the brightness temperatures with respect to the conservative variables, knowing the gradient with respect to the classical variables."

**Section 4.1**

• The background error cross correlations: I am not familiar with the variable  $(\theta_s)_a$ . Could you make a little clearer why you expect a link between  $(\theta_s)_a$  and  $q_t$  in clear-sky atmosphere (line 234)? If this should be evident from section 2.1 it would maybe be good to highlight it more.

This aspect is explained in the revised version by pointing to Equations (3) and (4) that show the linear relationship between  $(\theta_s)_a$  and  $q_t$ .

**Section 4.2**

• 1D-Var analysis fit to observations: In order to better compare the REF and EXP analyses in the regions of concern, it might be interesting to compute and compare RMSE and bias only within the blue rectangular boxes.

The statistics of the fit to the observation have been recomputed in the regions of concern corresponding to the blue rectangular boxes. We included the results in the new Table 2 and commented the results directly in the manuscript with the following sentences:

These statistics have also been calculated by restricting the dataset to the two dashed rectangular boxes represented in Figs 5 (b) and (c). A significant improvement is observed for channels the most sensitive to liquid water in the afternoon with a RMSE decreased from 4.3 K in the background to 0.57 K in REF and 0.37 K in EXP. For all the computed statistics, EXP always provides the best performance in terms of RMSE. Table 2 summarizes the bias and RMSE values obtained for the different samples.

**Conclusions**

• Line 300ff Maybe this paragraph would better suit to section 2.1, where the variables are introduced and described.

Indeed, this paragraph is moved into the section 2.1, as suggested.

• Line 326ff Do you expect results of TB to vary a lot after including q i in the conversion operator? Where there ice clouds in the presented case (line 327)?

We have added into the conclusion the following paragraph:

Indeed, inclusion of  $q_i$  in the conversion operator should lead to more realistic retrieved profiles of cloud condensates, and a 1D-var system with only  $q_l$  can create water clouds at locations where ice clouds should be present, as done in our experiment around 400 hPa between 15 and 24 UTC. However, since the frequencies of HATPRO are not sensitive to ice water content, the fit of simulated Tbs to observations could be reduced. In consequence, the synergy with an instrument sensitive to ice water clouds, such as a W-band cloud radar, is necessary to get improved retrievals of both  $q_i$  and  $q_l$ .

**4) Technical corrections**

**4.1) The main text:**

• Line 5: '[...] that are currently highly dependent on weather conditions [...]'  $\rightarrow$  '[...] that are highly

dependent on weather conditions when using classical variables, [...]'

The new text is introduced.

• Line 28: 'scheme'  $\rightarrow$  'schemes'

The name 'schemes' is used.

• Line 38: '[...] that data assimilation systems used to be [...]'  $\rightarrow$  '[...] that most data assimilation systems are [...]'

The new text is introduced.

• Line 67: Use  $s_{d0}(T_0, p_0)$  in the formula, as in line 69.

We have used  $s_{d0}(T_0, p_0)$  in the formula.

• Line 71: State that the explanation for  $\Lambda_r$  follows later in the section.

This sentence is added.

• Line 73: '[...] sublimation.'  $\rightarrow$  '[...] sublimation, respectively.'

The word 'respectively' is used.

• Line 106: Remove the point at the end of the title.

This is done.

• Line 143: 'set-up'  $\rightarrow$  'setup'

This change is made.

• Line 147: '[...] such a HATPRO [...]'  $\rightarrow$  '[...] such as a HATPRO [...]'

This change is introduced, together with a larger change suggested by the RC2.

• Line 155: 'lowest' instead of 'first few'

The word 'lowest' is introduced.

• Line 192: I propose to mention RTTOV-gb here after H because it was already mentioned earlier in the text.

The change is made.

• Line 195: comma after '(LWP)'

The comma is added.

• Line 201: No new paragraph before line 201

This 'new paragraph' is canceled.

• Line 236-237:  $B_z(\theta_s, q_t) \to B_z((\theta_s)_a, q_t)$ ?

This change is made (twice).

• Line 276: 'similarity'  $\rightarrow$  'similarities'

This change is done.

• Lines 284-287: Suggestion: 'Most of the liquid water is created in low clouds. Additionally, increments of  $q_l$  above 600 hPa are larger and more extended vertically and in time in EXP, where condensation occurs over a thicker atmospheric layer between 500 hPa and 300 hPa after 12 UTC. In the REF experiment, the creation of liquid water above 500 hPa only reaches values of 0.3 g/kg sporadically, for example at 21UTC. In the EXP setup, [...]'

We have introduced the proposed sentences.

• Line 289: '[...] keeps unchanged the vertical structure of the  $q_l$  profile [...]'  $\rightarrow$  '[...] keeps the vertical structure of the  $q_l$  profile unchanged [...]'

We have introduced your text.

• Lines 289-290: 'Liquid water is added where it already existed  $[...]' \rightarrow$  'In REF, liquid water is only added where it already exists [...]'

We have introduced your text.

• Line 293: '[...] show similar structures to [...]'  $\rightarrow$  '[...] show structures similar to [...]'

We have introduced your text.

• Line 296: 'value' instead of 'interest' ?

This change is done.

• Line 297: 'for assimilating'  $\rightarrow$  'to assimilate'

This change is done.

• Line 299: 'over South-Western France'  $\rightarrow$  'in South-Western France'

This change is done.

• Line 303: '(in e.g. fronts, [....])'

This change is done.

• Line 305: 'significance' instead of 'interest' ?

This change is done.

• Line 324: 'increment'  $\rightarrow$  'increments'

This change is done.

**4.2) The Figures:**

• Figure 1: Would it be possible to provide this figure with slightly larger font size?

The font of the annotations have been enlarged.

• Figure 2: – Do you mean 2 K instead of 0.2 K in the caption? – 'Relative Humidity' without capital letters – Could you add a description of the arrows (2(b), 2(d)) in the caption?

It is indeed 2 K; we write 'relative humidity'; we add:

"The vertical arrows in (b) and (d) indicate areas where potential temperatures are almost homogeneous or constant along the vertical."

• Figure 5: – Please make the numbering of the channels consistent between figure (1-13) and caption (0-12). Would it maybe be possible to even enumerate the channels in the figure after the MWR channel numbers (1-2,4-14 as channel 3 was removed, if I am correct)? – Provide units to the color bar. – I find it a little difficult to compare the values between panels (a) and (b)/(c) and really point out the magnitude of the reduction of deviations. Would it be possible to use the same range in the color bar of all three panels? Or alternatively at least point out the different scaling in the caption for clarity.

The same range of color bar cannot be used for the three panels because background departures have much larger values (up to 15 K) than analysis residuals (up to 3 K). However, the same scale has been adopted for the analysis residuals of the two 1D-Var systems. In order to avoid any visual misinterpretation of the plots, the color map is now different for the background departures and for the analysis residuals.

• Figure 6: I suggest centering the color map to fit 0 to the change of color between blue and red.

The color maps have been changed and centered to 0 between blue and red.

**Towards the use of conservative thermodynamic variables in data assimilation: preliminary results a case study using ground-based microwave radiometer measurements**

Pascal Marquet1, Pauline Martinet1, Jean-François Mahfouf1, Alina Lavinia Barbu1, and Benjamin Ménétrier2

[revised manuscript text omitted]

<sup>3Numerical Weather Prediction Satellite Application Facility

<sup>4Radiative Transfer for the TIROS Operational Vertical Sounder

---

## Author Comment (AC2)

**Answers to the Referee 2 about the paper amt-2021-361:**

"*Towards the use of conservative thermodynamic variables in data assimilation:*  **a case study** *using ground-based microwave radiometer measurements*"

by Pascal Marquet, Pauline Martinet, Jean-François Mahfouf,
Alina Lavinia Barbu, and Benjamin Ménétrier.

January 21, 2022

**1) Answers to the referee RC2**

• *The study analyzes the potential of introducing two moist-air conservables as control variables to a 1D var data assimilation system in order to improve short-term forecasts for fog conditions. A model setup with the new variables is compared to a conventional model setup based on a case study from the SOFOG3D campaign. New control variables require updated background error covariance matrices which are evaluated for stability under several meteorological conditions. The new setup is evaluated in model and observation space.*

• *The study is clearly written and the methodology is well described. The results are a good addition to the field and are well presented. The manuscript is convincing and I recommend publication after the following comments have been addressed.*

We thank you for this review and have taken into account almost all your comments and suggestions, as described below.

Please find, also, a document in "difference" mode where it is possible to isolate all changes in colour mode (from red to blue).

**2) Specific comments**

• *1) The aim of the study is generally clear. Yet, abstract, introduction and conclusion present different main scopes of this paper. I encourage the authors to align the main scope of the paper throughout the manuscript's sections.*

We thank the reviewer for pointing this inconsistency. We slightly modified the conclusion to be in line with the main scope of the paper,

**by changing the sentence:**

"*The aim of this study was to examine the interest of using moist-air entropy potential temperature $((\theta_s)_a)$ and total water content $q_t$ to study fog initiation and dissipation at small scale.*"

**into:**

"*The aim of this study was to examine the value of using moist-air entropy potential temperature $(\theta_s)_a$ and total water content $q_t$ as new control variables for variational assimilation schemes. In fact, the use of control variables less dependent to vertical gradients of $(T, q_v, q_l, q_i)$ variables should ease the specification of background error covariance matrices, which play a key role in the quality of the analysis state in operational assimilation schemes.*"

• *2) Ice-free conditions are assumed (line 97). Yet, Fig 2 reveals the presence of most probably ice-containing cirrus clouds in the analyzed case study. How is the resulting TB signal considered in the analysis and which errors are induced by the ice-free hypothesis not holding true?*

The inclusion of $q_i$ in the conversion operator should lead to more realistic retrieved profiles of cloud condensates. Indeed, a 1D-Var system with only $q_l$ can create water clouds at locations where ice clouds should be present, as done in our experiment around 400 hPa between 15 and 24 UTC. However,

since the frequencies of HATPRO are not sensitive to ice water content, the fit of simulated $TBs$ to observations could be reduced. In consequence, the synergy with an instrument sensitive to ice water clouds, such as a W-band cloud radar, is necessary to get improved retrievals of both $q_i$ and $q_l$. These statements are given in the conclusion of the revised version.

- *3) I find 'preliminary results' in the title very misleading. Which part of the shown analysis makes the study preliminary? I suggest to modify the title or to add an explanation to the conclusion section.*

The title has been modified as suggested by the RC1: "*Towards the use of conservative thermodynamic variables in data assimilation:* **a case study** *using ground-based microwave radiometer measurements.*"

- *4) Sec. 4.2: I am missing a comment on the performance of channel 8 in the analysis in Fig 5. Why does this channel show higher deviations in comparison to the other temperature sensitive channels? How are contributions by uncertainties in the forward model versus observational errors in the measured TB considered in the comparison?*

We have added the following explanations in the main text of the paper: :
"*We can note that the residuals are not as reduced for channel* 9 *(52.28 GHz) compared to other channels. Indeed, channels* 8 *and* 9 *(51.26 and 52.28 GHz) suffer from larger calibration uncertainties (Maschwitz et al 2013) and larger forward model uncertainties dominated by oxygen line mixing parameters (Cimini et al 2018) than other temperature sensitive channels. However, by comparing simulated brightness temperatures (TB) with different absorption models (Hewison et al 2007), or through a monitoring with simulated TB from clear-sky background profiles (DeAngelis 2017, Martinet et al 2020), larger biases are generally observed only at* 52.28 *GHz. Channel* 9 *is thus probably affected by additional instrumental biases compared to the other channels. Consequently, the higher deviations observed in figure* 5 *for channel* 9 *mostly originate from larger modelling and calibration uncertainties, which are taken into account in the assumed instrumental errors (prescribed observation errors of about* 3 *K for these two channels compared to* < 1 *K for other temperature sensitive channels) and also possibly from larger instrumental biases.*"

- *5) Sec. 4.3: I suggest adding another Figure to illustrate the characteristics of the Jacobians as given in the text.*

We have chosen not to present the Jacobians since the original aspect would be to examine them with respect to the new variables $(\theta_s)_a$ and $q_t$. The interpretation of these new Jacobians in $z$-space with respect to the ones obtained in $x$-space is non-trivial since it relies on the linearization of Eq. (7) to (10) with the distinction between "unsaturated" and "saturated" cases. This study has been published in an internal report, but we think that the additional equations, figures and text necessary for a thorough understanding are beyond the main focus of the paper. In order to partly address the reviewer's concern we have included in the revised version a reference to the paper of De Angelis et al. (2016) where the Jacobians in $x$-space are displayed (Figure 7).

- *6) Conclusion: I find the conclusions from the results sections 4.2 and 4.3 too short (lines 317-319). The added value of the new setup in particular for forecasts in foggy conditions should be highlighted as well as of the assimilation of TBs. The summary of the main results of both the comparison in model and observation space could be expanded by one or two additional sentences.*

The conclusion has been extended with the following statements:
"*The new 1D-Var has produced rather similar results in terms of fit of the analysis to observed TB values when compared to the classical one using temperature, water vapour and liquid water path. Quantitative results reveal slightly larger biases but smaller RMS values with the new system. On the other hand, the atmospheric increments are somewhat different in cloudy conditions between the two systems. For example, in the stratocumulus layer that formed during the afternoon, the new 1D-Var induces larger temperature increments and reduced liquid water corrections. Moreover, its capacity to generate cloud*

*condensates in clear-sky regions of the background has been demonstrated. As a preliminary validation, the retrieved profiles from the 1D-Var have been compared favourably against an independent observation data set (one radiosounding launched during the SOFOG3D field campaign). The new 1D-Var leads to profiles of temperature and absolute humidity slightly closer to observations in the planetary boundary layer.*"

**2) Technical Details**

- *Line 4: should read: SOFOG3D*

This typo had been fixed.

- *Line 106: no period needed in section title*

This is done

- *Line 113, 116: replaced weighted with 'weighed'*

We prefer to keep "weighted" because it is used in the sens of a "weighted sum" (https://en.wikipedia.org/wiki/Weight_function)

- *Line 147: should read "such as a HATPRO" (Humidity and Temperature PROfiler, Rose et al, 2005) instead of giving long title in line 187*

These changes are introduced.

- *Line 255: do the authors refer to the first seven channels instead of eight?*

We have changed the numbering of the channel in the main text and in the Figures according the new Table 1, with the channel 3 not studied and with the studied channels listed as: 1, 2 and then 4 to 14.

- *Line 289: sentence is unclear.*

Several sentences are modified, also due to RC1 suggestions.

- *Line 300: should read: motivation*

This change is done.

- *Lines 300-306: descriptions would better fit to introduction or Sec. 2*

These lines are moved into the section 2.1, as suggested.

- *Line 312: should read 'first' instead of 'firstly'*

This change is done.

**2) Figures**

- *Figs 3-6: add label and unit to color-bars*

We have added the units where they were missing in these figures.

- *Fig. 5: channel numbering should be as in text (1-13 (or 1-2; 4-14) instead of 0-12); explanation of the dashed blue lines should be added to figure caption. It would be desirable to have the analysis text closer to the figure to improve readability of the manuscript.*

- We have changed the numbering of the channel in the main text and in the Figures according the new Table 1, with the channel 3 not studied and with the studied channels listed as: 1, 2 and then 4 to 14.

- The dashed blue boxes are explained.

- New analysis and explanations have been added, with in particular a new color-bar for the first figure which is different from the color-bars for the two other figures.

[revised manuscript text omitted]

---

## Author Response (AR2)

**Answers to the Editor and Referee 2 about the paper amt-2021-361:**

"*Towards the use of conservative thermodynamic variables in data assimilation: a case study using ground-based microwave radiometer measurements*"

by Pascal Marquet, Pauline Martinet, Jean-François Mahfouf,
Alina Lavinia Barbu, and Benjamin Ménétrier.

March 7, 2022

Dear Editor and Reviewer-2

I was able to modify the article with my co-authors in order to take into account all the requests for modifications concerning:

 1) the units + legends of axes and colorbars of figures 3, 4, 5 and 7;

 2) the caption of the figure 3;

 3) the use of "University of Cologne" (2 times) in the acknowledgments.

We provide a "latexdiff" pdf file in order to show (in red) the changes in the caption of the figure 3 and in the acknowledgments, together with the new figures 3, 4, 5 and 7 included.

We provide a zip file with original resolution and lower resolution (LR) figures, when available:

Fig1_R1.jpg    Fig1_R1_LR.jpg
Fig2_R1.jpg    Fig2_R1_LR.jpg
Fig3_R2.jpg
Fig4_R2.jpg
Fig5_R2.jpg
Fig6_R1.jpg    Fig6_R1_LR.jpg
Fig7_R2.jpg

Best regards,

 Dr.Hab. Pascal Marquet

[revised manuscript text omitted]